# An Approach for Modelling Harnesses in the Extreme near Field for Low Frequencies

Anargyros T. Baklezos [1], Theodoros N. Kapetanakis [1], Ioannis O. Vardiambasis [1], Christos N. Capsalis [2] and Christos D. Nikolopoulos [1,*]

1 Department of Electronic Engineering, Hellenic Mediterranean University, GR-73133 Chania, Greece; abaklezos@hmu.gr (A.T.B.); todokape@hmu.gr (T.N.K.); ivardia@hmu.gr (I.O.V.)
2 School of Electrical and Computer Engineering, National Technical University of Athens, GR-15780 Zografou, Greece; ccaps@central.ntua.gr
* Correspondence: cnikolo@hmu.gr

**Abstract:** A key part of every space science mission, in the system-level approach, is the detailed study and modeling of the emissions from transmission lines. Harnesses usually emit electromagnetic fields due to the currents (of common and/or differential modes) that flow on their shields. These fields can be identified via conducted emissions measurements. Relying on the operating frequency, any cable can be considered as a dipole or a traveling-wave antenna. Limited work can be found in the literature regarding modeling methodologies for cable topologies, especially in the low frequency (ELF, SLF, VLF, LF) domain. This work intends to provide perceptions for the precise estimation of harness radiated emissions, consider a mission-specific measurement point (where the sensors are placed), and follow ESA's recent science mission studies for electromagnetic cleanliness applications. For the low frequencies considered herein, any linear cable path is considered as a point source (infinitesimal dipole) and we evaluate its effect on the calculated electric field extremely close to the source. For such distances, it is shown that the dipole representation is not accurate. To remedy this phenomenon, this article proposes a methodology, which can be easily expanded to complex cable geometry cases.

**Keywords:** low-frequency harness model; Hertzian dipole; near-field approximation; source segmentation

## 1. Introduction

The routing of long harnesses with lengths of a few meters is a common issue for EMC engineers, and in the literature, this can be found in a lot of studies addressing the majority of the problems generated by complex geometries and long paths. Indicatively, Mora et al. [1] present an MTL model for a spacecraft harness regarding the shielding performance of a multiconductor cable with a braided shield, while Arianos et al. [2] and Ridel [3] identify the need for characterizing and modeling complex harness structures. However, limited research has been conducted when a low-frequency field evaluation is required under strict cleanliness requirements regarding the measurement sensor's position (e.g., in the THOR space mission, the EFI-HFA sensor on the spacecraft's boom was placed at approximately 6 m away) [4,5]. In the single dipole instance, for an operational frequency between 0 and 200 kHz, a harness of several meters can be safely attributed as a point source when compared to the corresponding wavelength of several kilometers. However, when the distance between the source and the measuring sensor is similar to the length of the cable, the infinitesimal dipole's approximation, referred to in antenna theory [6], is not valid. To overcome this problem, one approach could be the splitting of the long harness into very small Hertzian dipole segments, fulfilling the condition. This work investigates the necessity of source segmentation, depending on the ratio of the observatory distance to the cable length; an idea originally presented in [7]. It should be noted that the proposed analysis can be also applied in more complex multiconductor cable geometries, considering

the current flow upon the shield as the electromagnetic radiation source. Following the aforementioned approach, the common-mode (CM) and differential-mode (DM) currents, identified during the measurement campaign of conducted emissions, can be treated separately as individual sets of various dipole sources. Consequently, conducted emissions can be used to predict radiated emissions, since the only parameter needed is the CM or DM current distribution.

## 2. Near-Field Approximations Very Close to the Dipole Source

### 2.1. Near Field Representation of Harnesses in the Low-Frequency Domain

In the low-frequency domain, the electric length of a typical harness with straight path geometry (up to some meters) is a minuscule fraction of the wavelength, thus its radiation can be approximated by an infinitesimal dipole for most intents and purposes. It should be noted, that for frequencies below 200 kHz, and wavelengths larger than 1.5 km, any current (CM or DM) in a route of several meters can be assumed constant. In general, the electric field of a Hertzian dipole is calculated by applying the following equations [6]:

$$E_r = \eta \frac{I_o l cos\theta}{2\pi r^2} \left[ 1 + \frac{1}{jkr} \right] e^{-jkr} \tag{1}$$

$$E_\theta = j\eta \frac{k I_o l sin\theta}{4\pi r} \left[ 1 + \frac{1}{jkr} - \frac{1}{(kr)^2} \right] e^{-jkr} \tag{2}$$

$$E_\varphi = 0 \tag{3}$$

In real cases, harness routing is not limited to single straight lines, but it follows complex geometries. In order to account for such cables, we propose the division of the complex geometry into the minimum number of necessary straight paths, and the representation of each of them as a Hertzian dipole centered in the corresponding path.

An indicative geometry showcasing the proposed rationale is depicted in Figure 1. The length of each segment dipole is equal to one of the corresponding straight paths (e.g., segments OA, AB, BC, CD).

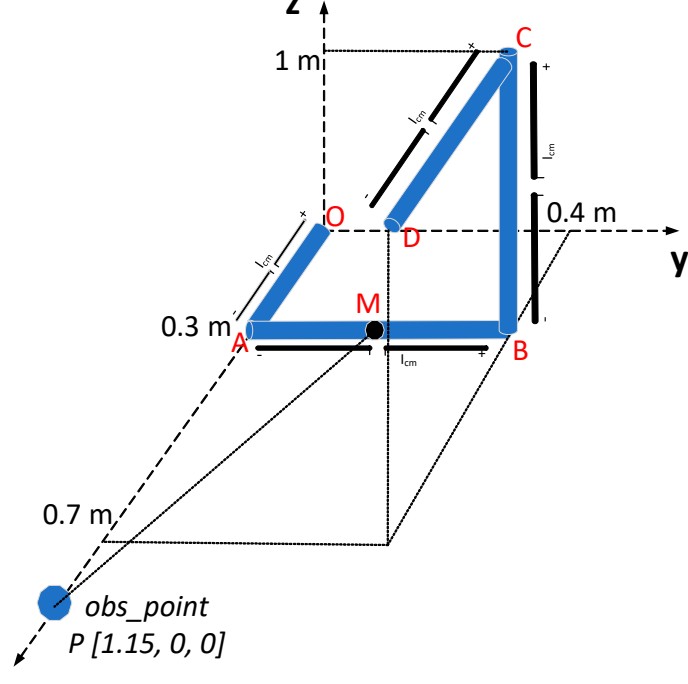

**Figure 1.** Application of the near-field dipole segmentation technique in a complex cable geometry.

For cleanliness purposes, in the case of spacecraft sensors and other sensitive payloads, their location is considered as the observation point P, at which the emissions can be calculated as the superposition of the electric field of each segment dipole. Obviously, in the low-frequency domain, the near-field approximation is well justified, so we use Equations (4)–(6) ($kr \ll 1$, $kr \ll \lambda$, $r \ll 2\pi$) [6] in order to calculate the fields, instead of the general Equations (1)–(3).

$$E_r \cong -j\eta \frac{I_0 l e^{-jkr}}{2\pi kr^3} cos\theta \tag{4}$$

$$E_\theta \cong -j\eta \frac{I_0 l e^{-jkr}}{4\pi kr^3} sin\theta \tag{5}$$

$$E_\varphi = 0 \tag{6}$$

In order to carry out the superposition in Cartesian coordinates, the electric field components that are given by Equations (1)–(3) or (4)–(6) in spherical coordinates need to be converted according to the following transformations [6]:

$$E_x = E_r sin\theta cos\varphi + E_\theta cos\theta cos\varphi - E_\varphi sin\varphi$$

$$E_y = E_r sin\theta sin\varphi + E_\theta cos\theta sin\varphi + E_\varphi cos\varphi \tag{7}$$

$$E_z = E_r cos\theta - E_\theta sin\theta$$

The above electric field component expressions are valid at every point, except from the source itself [6]. The main condition for this formulation to hold is that the source length has to be electrically small in comparison to the wavelength, which is valid for the harness case under study. However, another condition that has to be also fulfilled is that the source has to be short enough in order for the distance between a random source point and an arbitrary observation point P to be identical to the distance between the source center and P. In the case of a sensor (observation point P) located near the spacecraft (and to the cable also), their distance is comparable to cable's length, resulting in that the second condition does not hold true and extra steps are necessary to accurately calculate the harness emissions.

Equations (1)–(3) and (4)–(6) are valid when (i) the wire is very short (with length $l \ll \lambda$) and (ii) very thin (with diameter $\alpha \ll \lambda$), while (iii) the distance $r$ between any point on the source and the observation point can be considered constant [6]. This last condition for $r$ does not hold true when the observation point is very close to the source, which is the case under study. In order to overcome this, we need to consider the general problem of multiple infinitesimal sources carrying electric currents and individually satisfying each of the above three necessary conditions. The vector potential $A(x, y, z)$ is calculated as the sum of the individual vector potentials of all sources:

$$A(x,y,z) = \sum A_i(x,y,z) = \sum \left( \frac{\mu}{4\pi} \int I_{e,i}(x'_i,y'_i,z'_i) \frac{e^{-jkr_i}}{r_i} dl'_i \right) \tag{8}$$

where $I_{e,i}(x'_i,y'_i,z'_i)$ is the current of the source $i$, $r_i$ is the distance between any point of the source $i$ and the observation point $(x,y,z)$, and $dl'_i$ is the length of the source $i$. It should be noted that in the case of a single harness topology with the same current, the current is constant for every $i$ ($I_{e,i}(x'_i,y'_i,z'_i) = I_0$). Consequently, the total field at the observation point is the sum of the fields of all individual sources according to (1)–(3). This is valid for an infinite number of segments, but taking into account a specific required level of accuracy, it can be finally approximated with a finite number of segments. Thus, by carefully choosing the proper segmentation of the source—in order for its geometrical characteristics to fulfill the three necessary conditions—the radiated field can be estimated with satisfying accuracy. In the next section, we showcase the aforementioned segmentation technique and discuss its applicability.

### 2.2. Considerations for Observation Distances Comparable to the Cable's Length

In order to investigate the applicability of Equations (1)–(3) or (4)–(6), when the observation distance is comparable to the source dimensions (harness length), we have implemented the following specific scenario: a radiating source consisting of a 2 m straight cable is fed with a 1 A current oscillating with a 9 kHz frequency. For a 2 m indicative observation distance, the second condition is not valid, so the cable cannot be considered as a single point source; however, after proper segmentation, the second condition holds true for each of the segments.

If the number of source segments (Hertzian dipoles) is $N$, the total field radiated from the cable assembly at the observation point P is given by:

$$E_{cable} = \sum_{i=1}^{N} E_i \tag{9}$$

The parameters of this segmentation are: (i) the number $N$ of the segment dipoles that make up the whole cable, and (ii) the distance r from the center of the cable to the observation point P. In order to take into account the length $L$ of the cable as well, we define and investigate the $r/L$ ratio. Thus, the length itself is not a standalone variable anymore, and our study focuses on the relative to cable length distance.

In order to showcase the effect of the parameter *ratio* along with the segmentation number $N$, we have investigated the following four indicative cases:

1.  Single Dipole Case: is when the electric field is evaluated from Equations (1)–(3), considering that the source (cable) is one dipole with length $L$ equal to the cable length.
2.  Segmented Cable Case: is when the electric field is evaluated from the superposition of the electric fields of $N$ segment dipoles, each has a length equal to $L/N$ laying consecutively on the cable path with its center at $-L/2 + L/2N + i*L/N$ ($i = 0, \ldots, N-1$), and contributing to the total field with its segment field calculated from (1)–(3).
3.  Single Dipole Case with Near Field Approximation: is when the electric field is evaluated from Equations (4)–(6), considering that the source (cable) is one dipole with length L equal to the cable length.
4.  Segmented Cable Case with Near Field Approximation: is when the electric field is evaluated from the superposition of the electric fields of $N$ segment dipoles, each having a length equal to $L/N$, laying consecutively on the cable path with its center at $-L/2 + L/2N + i*L/N$ ($i = 0, \ldots, N-1$), and contributing to the total field with its segment field calculated from (4)–(6).

We have studied the configuration of Figure 2 for several values of the elevation angle $\theta$, but all results presented herein correspond to $\theta = 90°$, for which segmentation of the source is needed for a larger distance, corresponding to *ratio* = 4.

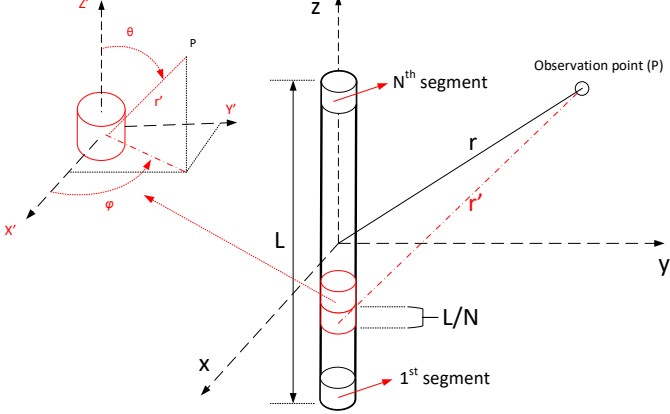

**Figure 2.** Dipole segmentation for near-field approximation study.

For the extreme near-field calculations of the complex values of the electric field component $E_x$, $E_y$, or $E_z$, the imaginary part dominates over the real part, and its calculation convergence (in the Segmented Dipole Case) may be achieved for values of $N$ higher than 20 segments when the *ratio* value is 0.8.

As discussed earlier in Section 1, theoretically, an infinite number of infinitesimal sources are necessary for the accurate evaluation of the total electric field. However, practically, a sufficiently high number of segments can be used to approximate the total electric field without a significant loss (<1%) in the accuracy. In order to estimate the minimum number required of segments for an as accurate as possible (within acceptable limits) calculation of the total electric field, we have investigated the convergence point of the electric field for several values of $N$ and *ratio*.

Figure 3 reveals the impact of segmentation in the calculation of the electric field amplitude for $N$ = 5, 50, and 50,000 segments. It is clear, however, that segmentation has a major impact for values of *ratio* below 4, while for values of *ratio* above 4, the Single Dipole Case and the Segmented Cable Case (for $N$ = 50,000) converge. Moreover, Figure 4 suggests that significantly fewer segments ($N$ = 50) are actually enough to achieve an accuracy better than 99.9%.

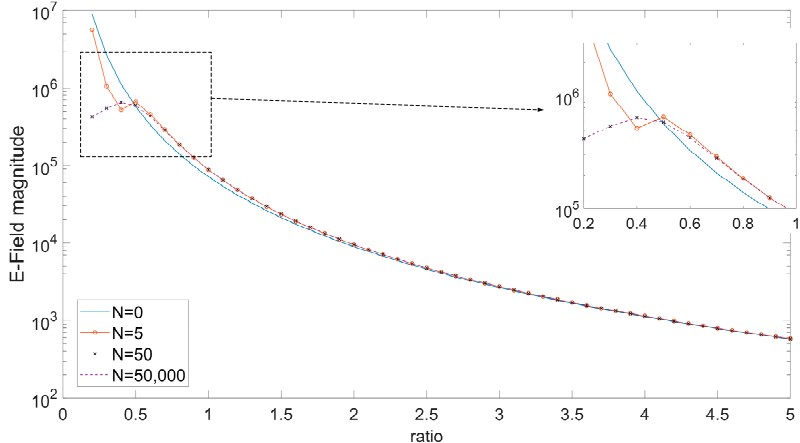

**Figure 3.** Comparison of the magnitude of the electric field $(E_x^2 + E_y^2 + E_z^2)^{1/2}$ versus the *ratio* of the measurement distance to the cable length, for $N$ = 0 (straight blue line), 5 (bulleted red line), 50 (x marks), and 50,000 (dashed line) segments.

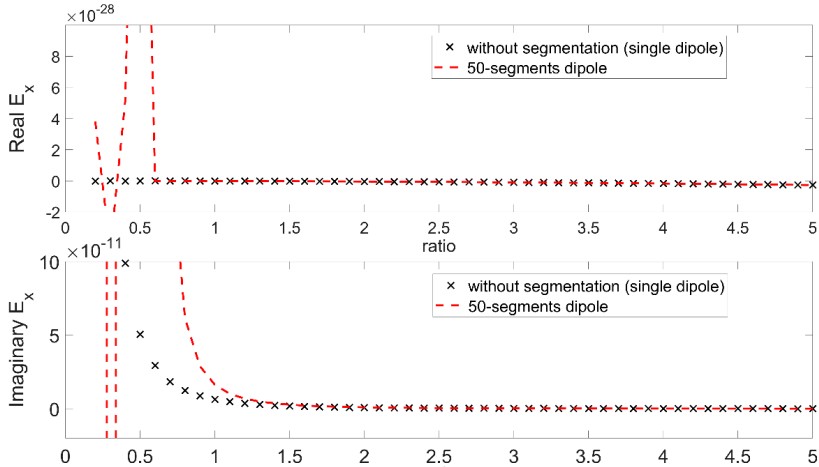

**Figure 4.** Comparison of the real and imaginary parts of the electric field component $E_x$ versus the *ratio* of the measurement distance to the cable length, for the Single Dipole Case (x marks) and the Segmented Dipole Case for $N$ = 50 segments (dashed lines).

　　　Accordingly, the real part of the electric field is many orders smaller than the imaginary, thus all the analysis is based on the imaginary part (which is dominant in magnitude). Thus, in this case, the segmentation is mandatory and the field result differentiates from the Single Dipole Case where one dipole is considered. As a result, $N = 50$ is considered a sufficiently safe choice for the ratio parameter evaluation.

　　　Figures 4–6 showcases the comparison between the Single Dipole Case and the corresponding Segmented Dipole Case with $N = 50$ subparts. This comparison has been performed for different values of the *ratio* parameter to identify the minimum observation distance (compared to the cable length) for which the source segmentation is not required anymore. Moreover, the calculations make apparent that for distances four times larger than the cable length (*ratio* > 4), the difference between the Single Dipole Case and the Segmented Cable Case is less than 5%. This percentage is inversely proportional to the *ratio*, so it decreases when the *ratio* increases. The same remarks also apply to the $E_y$ and $E_z$ electric field components.

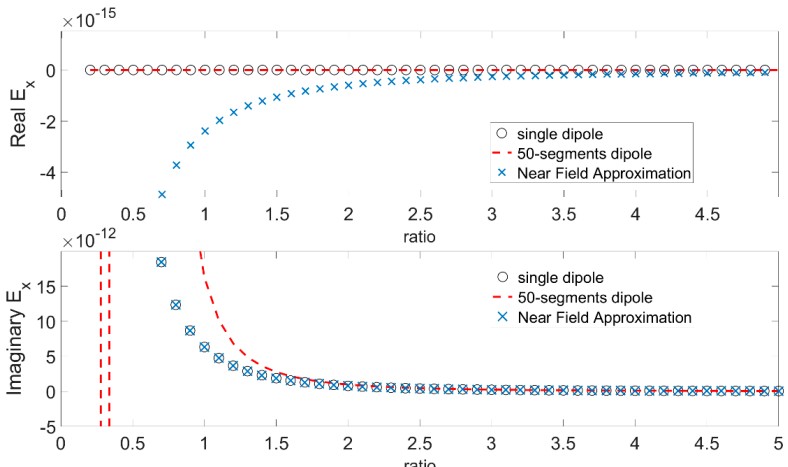

**Figure 5.** Comparison of the real and imaginary parts of the electric field component $E_x$ versus the *ratio* of the measurement distance to the cable length, for the Single Dipole Case (circles), the Segmented Dipole Case for $N = 50$ segments (dashed lines), and the Single Dipole Case with Near Field Approximation (x marks).

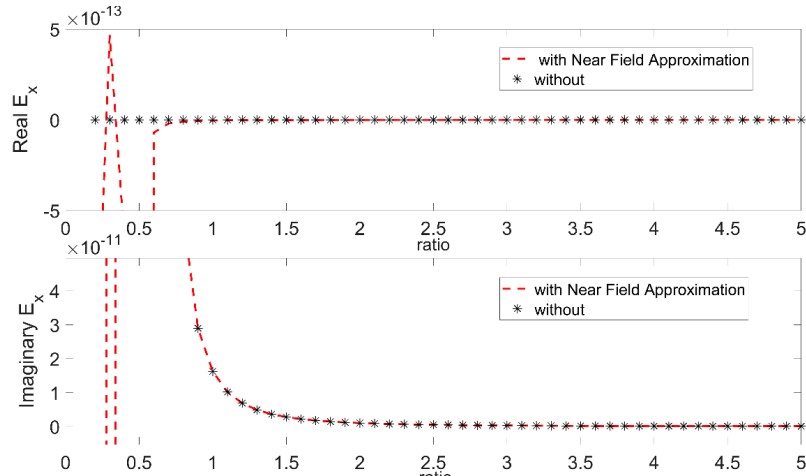

**Figure 6.** Comparison of the real and imaginary parts of the electric field component $E_x$ versus the *ratio* of the measurement distance to the cable length, for the Segmented Cable Case (star marks) and the Segmented Cable Case with Near Field Approximation with $N = 50$ segments (dashed lines).

Figure 3 reveals that the convergence of the electric field calculation is obtained for segmentation numbers *N* less than 50. In fact, the field convergence is improved by less than 0.1% when *N* increases from 20 to 50. However, for the rest of the cases presented herein, and in order to maximize the accuracy, we have used *N* = 50 segments.

The above considerations have also been confirmed for the 90 kHz frequency. Additional calculations, which were carried out considering different observation distances (r = 4 m and r = 8 m) but with the same *ratio* values, have verified the aforementioned conclusions. Thus, the current study proposes a practical rule for modeling harnesses either with a single Hertzian dipole or with a set of multiple Hertzian dipoles arising after the appropriate segmentation process, when the observation point is in close proximity to the source.

### 3. Application of the Segmentation Technique in Complex Geometries

As a means for exhibiting the application of the herein proposed technique in a more practical and complex cable geometry, the scenario depicted in Figure 1 is investigated. A cable with a 2.1 m total length is fed with a 25 μA current of 9 kHz frequency and routed across the three planes depicted in Figure 1. In order to decide what is the appropriate distance for the single dipole representation, or if the segmentation technique is necessary for the complex cable geometry, a cable's reference point has to be selected for the calculation of the *ratio* parameter. Based on the methodology of Section 2, the mid-point of the nearest to the observation point segment is the best candidate for the cable's reference point. Thus, the distance from this reference point to the observation point effectively determines the *ratio* parameter. The necessary steps, in order to decide on the appropriate modeling strategy, are summarized in the following pseudo-code:

Input (observation point coordinates)
Input (cable segments start/endpoint coordinates)
Input (current distribution)
For *i* = 1 to number of segments
Calculate mid-point coordinates for segments *i*,
Calculate mid-point—observation point distance
end
set reference point coordinates equal to the coordinates of the mid-point with the minimum distance.
Calculate ratio parameter.

In the complex under investigation geometry of Figure 1, the nearest mid-point (M of Figure 1) is that of the second segment with (*x*, *y*, *z* = 0.3, 0.2, 0.0), whose distance from the selected observation point P is 0.8732 m. For a cable with a length of 2.1 m in total, the value of the ratio becomes 0.4158, which dictates the selection of the segmentation technique. It should be noted that when the electric field magnitude at the (under consideration) observation point is 12.1 V/m when the segmentation technique is used, and 17.5 V/m without the segmentation, this yields a difference of 44.6%.

Furthermore, Figures 7 and 8 present the deviation percentage of the field calculated under the two modeling approaches (with and without segmentation) with respect to the *ratio* parameter. The impact of the segmentation technique is apparent from the comparison between the two cases, since the deviation of the field magnitude ranges from 104% (for *ratio* = 0.04) to 1.6% (for *ratio*= 5). This deviation justifies the use of the proposed segmentation technique in order to more accurately model a harness in extremely low frequencies and especially in close proximity to the cable geometry.

Moreover, Figure 8 shows that the near-field approximation used in Cases 3 and 4 becomes valid with less than 5% discrepancy for *ratio* values greater than 2.

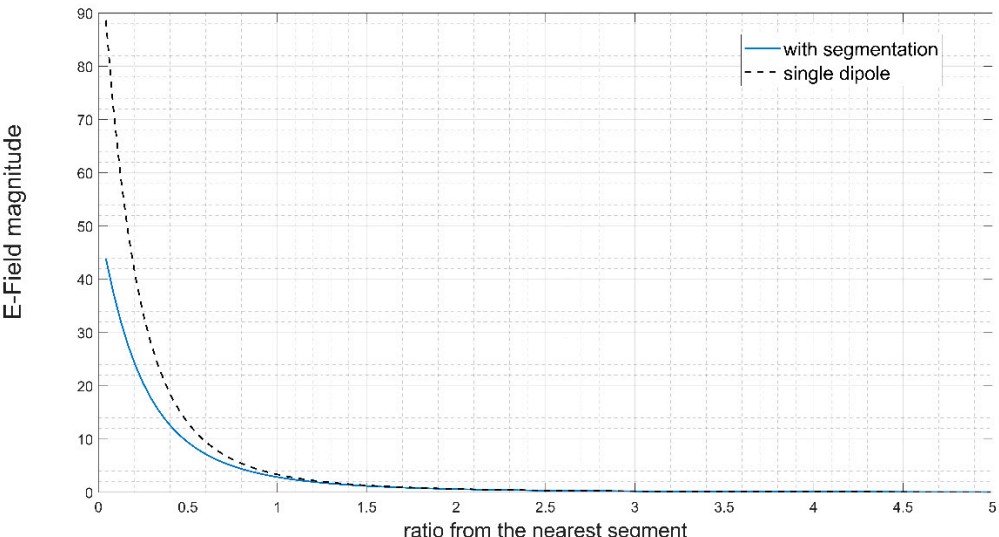

**Figure 7.** Comparison of the magnitude of the electric field $(E_x{}^2 + E_y{}^2 + E_z{}^2)^{1/2}$ versus the *ratio* of the measurement distance to the cable length, for the Single Dipole Case (dashed line) and the Segmented Cable Case with *N* = 50 segments (solid line).

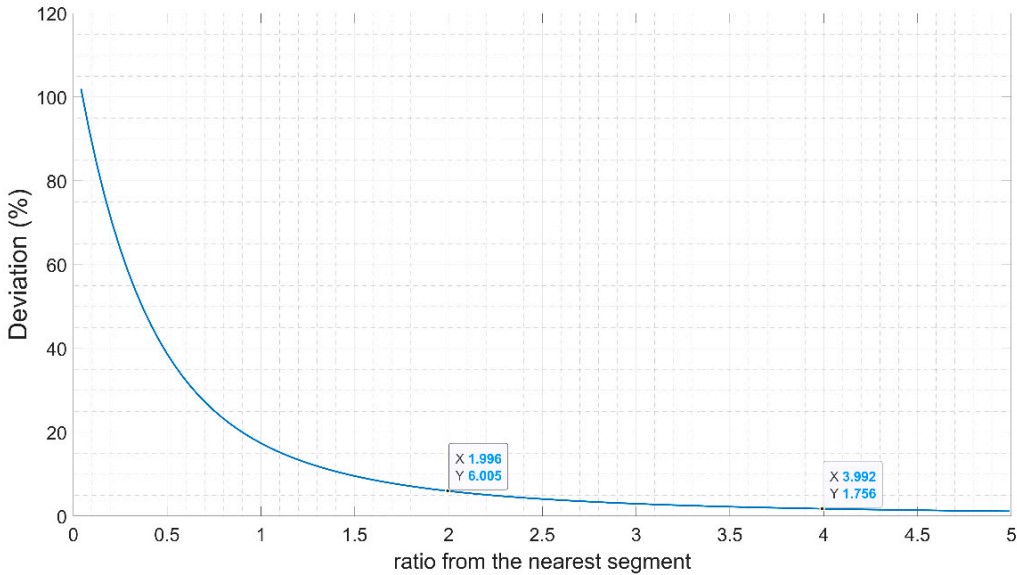

**Figure 8.** Deviation percentage between the two modeling approaches.

## 4. Conclusions

In this work, we evaluate the accuracy of low-frequency electric field calculation, in cases where the length of the harness is comparable to the observation distance. For the accurate calculation of the field extremely close to the source, we propose the segmentation of the long harness. Furthermore, 20 segments are sufficient, since increasing their number up to 50 does not improve the field calculation more than 0.1%. Moreover, when the ratio of the measurement distance to the harness length is greater than 4, the harness can be approximated by a Hertzian dipole, since the field produced with and without segmentation differs less than 2%. The proposed methodology was applied in a typical complex cable geometry straightening, as per the authors' claims. This work has specific merit on space cleanliness applications, where unintentional harness emissions can cause interference to sensitive measuring sensors.

**Author Contributions:** Conceptualization, A.T.B. and C.D.N.; methodology, A.T.B. and C.D.N.; software, A.T.B., T.N.K. and C.D.N.; validation, A.T.B., T.N.K. and C.D.N.; formal analysis, A.T.B., I.O.V. and C.D.N.; investigation, A.T.B., I.O.V. and C.D.N.; resources, A.T.B, I.O.V. and C.D.N.; data curation, A.T.B., T.N.K. and C.D.N.; writing—original draft preparation, A.T.B. and C.D.N.; writing—review and editing, A.T.B., I.O.V. and C.D.N.; visualization, A.T.B., I.O.V. and C.D.N.; supervision, C.D.N. and C.N.C.; project administration, C.D.N. and C.N.C. All authors have read and agreed to the published version of the manuscript.

**Funding:** This research received no external funding.

**Data Availability Statement:** The data used in this study are available on request from the corresponding author. The data can be easily reproduced from the theoretical analysis described in the study.

**Conflicts of Interest:** The authors declare no conflict of interest.

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
