# Peer review of "An Approach for Modelling Harnesses in the Extreme near Field for Low Frequencies"

_applsci, doi:10.3390/app12063202_

Round 1

Reviewer 1 Report

The authors  evaluate the accuracy of low-frequency electric field calculation in cases where the length of the harness is comparable to the observation distance, and proposed the segmentation of a long harness for accurate field calculation

The main drawback is the lack of experimental validation or theoretical derivation of the accuracy of the new method.

Reviewer 2 Report

This work proposes accurate modeling of wiring emissions by considering a straight cable path as an infinitesimal dipole source when modeled in the low-frequency region and investigates the implications on the resulting electric field in extreme proximity to the source.

English usage is not good

Reviewer 3 Report

It is necessary to review the whole representation of the problem as it is now in the equations. There is not a proper relationship between figures and variables, furthermore it is necessary a clear explanation of each variable in order to avoid clear misunderstandings. The proposed experiments are quita valuable but itself need to be extensively reviewed  before accepting. 

Round 2

Reviewer 3 Report

I recommend  to review the style of this manuscript as well as the results. 
